# Effects of Liposomal Vitamin C, Coenzyme Q10, and Bee Venom Supplementation on Bacterial Communities and Performance in Nile Tilapia (*Oreochromis niloticus*)

**DOI:** 10.3390/biology14030309

**Published:** 2025-03-19

**Authors:** Islam I. Teiba, Yasser S. A. Mazrou, Abeer H. Makhlouf, Nabil I. Elsheery, Sahar Hussein Abdalla Hekal, Nermeen M. Abu-Elala, Mahmoud Kamel Bakry, Emad H. El-Bilawy, Akram Ismael Shehata

**Affiliations:** 1Department of Agricultural Botany, Faculty of Agriculture, Tanta University, Tanta 31527, Egypt; 2Business Administration Department, Community College, King Khalid University, Guraiger, Abha 62529, Saudi Arabia; ymazrou@kku.edu.sa; 3Department of Agricultural Botany, Faculty of Agriculture, Minufiya University, Shibin El-Kom 32511, Egypt; hyabeer@yahoo.com; 4Department of Natural Resources, Faculty of African Postgraduate Studies, Cairo University, Giza 12613, Egypt; saharhekal@cu.edu.eg; 5Department of Aquatic Animal Medicine and Management, Faculty of Veterinary Medicine, Cairo University, Giza 12613, Egypt; nermeen_abuelala@cu.edu.eg; 6King Salman International University, Ras Sedr Campus, South Sinai 46618, Egypt; mahmoud.kamel@ksiu.edu.eg (M.K.B.); emad.elbilawy@ksiu.edu.eg (E.H.E.-B.); 7Department of Animal and Fish Production, Faculty of Agriculture (Saba Basha), Alexandria University, Alexandria 21531, Egypt; akramismael2@alexu.edu.eg

**Keywords:** antimicrobial activity, gut microbiota, hepatointestinal histology, immune response, nutraceuticals, *Oreochromis niloticus*

## Abstract

This study examined the effects of liposomal vitamin C, CoQ10, and bee venom on Nile tilapia health and performance. The 60-day feeding trial investigated impacts on growth, digestive enzymes, gut microbiota, tissue histology, immune function, and antioxidant responses. All nutraceuticals significantly improved these parameters, with bee venom demonstrating the most pronounced effects. Fish supplemented with these compounds showed enhanced digestive enzyme activities, beneficial gut microbiota shifts, improved tissue morphology, elevated immune markers, and increased antioxidant responses. The findings suggest the potential benefits of incorporating these nutraceuticals into tilapia aquaculture diets to optimize fish health and performance.

## 1. Introduction

Aquaculture is a rapidly growing sector of global food production, driven by the increasing demand for high-quality, protein-rich aquatic products [1]. However, the intensification of aquaculture practices is accompanied by significant challenges, including suboptimal growth performance, compromised immune function, and oxidative stress [2,3]. These factors can negatively impact fish health and productivity, ultimately limiting the sustainability and efficiency of aquaculture systems [4]. In response to these challenges, there has been growing interest in the use of natural nutraceuticals as functional dietary supplements to enhance fish health, resilience, and overall performance [5,6]. Nutraceuticals such as liposomal vitamin C (VC), coenzyme Q10 (CoQ10), and bee venom (BV) have garnered attention for their potential to improve various physiological parameters in aquaculture species [7,8,9].

L-ascorbic acid (VC) is a water-soluble micronutrient essential for numerous physiological processes in aquatic organisms, including antioxidant defense, collagen synthesis, and immune modulation [10]. Unlike terrestrial animals, most fish species, including Nile tilapia, lack the enzyme L-gulonolactone oxidase, which is required for the endogenous synthesis of VC, making dietary supplementation essential [11]. Conventional VC supplementation in aquaculture, however, faces significant challenges due to its instability under environmental conditions such as high temperature, oxidative stress, and extended storage, which reduce its bioavailability and efficacy [12]. To address these limitations, advanced nutrient delivery systems, such as liposomal encapsulation, have been explored. Liposomes, spherical vesicles composed of phospholipids, enhance the stability and bioavailability of bioactive compounds by protecting them from degradation and facilitating their uptake [13]. Recent studies have demonstrated that VC can improve growth performance, antioxidant capacity, and immune responses in aquaculture species [14]. Moreover, previous studies have investigated the optimal dietary VC requirements for various aquaculture species, including *Myxocyprinus asiaticus*, *Pelteobagrus fulvidraco*, and *Rachycentron canadum* among others [15,16,17].

Coenzyme Q10 (CoQ10) has also gained attention as an important functional nutrient in aquaculture. It is a lipid-soluble antioxidant and a critical component of the mitochondrial electron transport chain, where it plays a vital role in cellular energy production and defense against oxidative stress [18]. Dietary supplementation with CoQ10 has been shown to enhance growth performance, feed conversion efficiency, and immune responses in various fish species, highlighting its potential as a functional feed additive in aquaculture [19,20]. Additionally, CoQ10 supplementation has been found to alleviate oxidative stress and mitigate the adverse effects of heavy metal exposure in fish, further emphasizing its role in improving fish resilience under intensive farming conditions [21].

Bee venom (BV) is a natural substance that has received attention for its potential benefits in aquaculture. Containing bioactive compounds such as melittin, apamin, and phospholipase A2, it exhibits potent anti-inflammatory, antimicrobial, and antioxidant properties [22]. These bioactive molecules contribute to BV’s immunomodulatory effects, which have shown promise in enhancing disease resistance and overall health in both livestock and aquaculture species [23,24,25,26]. Recent studies have suggested that BV supplementation can improve tissue histology, gut microbiota composition, and antioxidant responses, offering a novel approach to improving aquaculture performance [8].

Nile tilapia (*O. niloticus*) is one of the most widely cultured freshwater fish species globally, recognized for its rapid growth, high feed efficiency, and adaptability to diverse environmental conditions. Representing approximately 10% of global aquaculture production, it plays a critical role in food security and income generation, particularly in developing countries [27]. Despite its economic importance, the intensive farming of Nile tilapia often exposes the fish to various stressors, including poor water quality, high stocking densities, and disease outbreaks [28]. These stressors can compromise fish health and performance, leading to suboptimal growth, weakened immune function, and increased susceptibility to disease. Addressing these challenges through innovative nutritional interventions is crucial for enhancing the efficiency and sustainability of tilapia farming systems [29].

Although individual studies have investigated the effects of liposomal VC, CoQ10, and BV supplementation on fish health and performance, comparative research evaluating their relative impacts on aquaculture species, particularly Nile tilapia, remains scarce [8,30,31]. Additionally, there is a limited understanding of how these nutraceuticals influence critical physiological parameters, such as gut microbiota composition, digestive enzyme activity, and the histological features of vital organs. Addressing these knowledge gaps necessitates an integrated research approach that evaluates the effects of advanced delivery systems and nutraceuticals on the overall health and productivity of farmed fish. Therefore, the present study aims to compare the effects of liposomal VC, CoQ10, and BV on growth performance, immune function, gastrointestinal digestive enzyme activity, hepatointestinal histology, and antioxidant responses in Nile tilapia. By examining these natural nutraceuticals comparatively, this research seeks to provide valuable insights into their potential applications for enhancing the efficiency and sustainability of tilapia farming systems, while identifying the most effective supplementation strategies for promoting optimal fish health and productivity.

## 2. Materials and Methods

### 2.1. Ethics Approval Statements

The Ethics and Animal Welfare Committee of the Faculty of Desert Agriculture, King Salman International University, Egypt, approved all experimental procedures, including fish maintenance and handling, under Approval No. KSIU/2024/DA-9. These procedures were performed in strict accordance with the ARRIVE guidelines version 2.0.

### 2.2. Experimental Setup and Laboratory Conditions

The feeding trials were conducted at the Faculty of Desert Agriculture, King Salman International University, Egypt (29°35′21.5″ N 32°46′10.3″ E). Nile tilapia (*O. niloticus*) juveniles were obtained from a private fish farm in Kafr Elsheikh City, Egypt. The experimental system consisted of a 1000 L cylindrical fiberglass tank for acclimatization and twelve 200 L fiberglass experimental tanks equipped with continuous aeration. Fish were acclimatized for 14 days prior to the feeding trial and fed a basal diet (Table 1). Following acclimatization, fish with an initial average weight of 35.17 ± 0.22 g were randomly distributed into experimental tanks at a density of 30 fish per tank, with three replicates per treatment.

Throughout the 60-day trial, water quality parameters were monitored using digital meters. Dissolved oxygen levels (5–7 mg/L) were measured with a DO meter (Hanna, Romania, Model HI 9147), while pH (6.5–7.5) was assessed using a pH meter (Model WT-80). Temperature (26–28 °C) was maintained using a digital thermometer. A 12:12 h light: dark photoperiod was maintained to simulate optimal conditions.

### 2.3. Supplement Sources, Diet Preparation, and Procedure

Liposomal vitamin C (LVC, ASIN: B00JFF48I6) was commercially sourced, while CoQ10 was obtained from the MEPACO Company, Cairo, Egypt (purity > 98%) [30]. Bee venom was harvested utilizing a semiautomatic collection system consisting of electrified stainless-steel electrodes connected to a battery and pulse generator assembly with an integrated glass slide. The application of electrical stimuli (Voltage: 11.5-13.5 VDC; Timer ON: 0.5–2 s; Timer OFF: 3–5 s) induced stinging behavior, causing bees to deposit venom onto the glass surface. As the volatile components naturally evaporated, the venom solidified into a white precipitate, which was carefully scraped off the glass surface for collection. The purified venom was stored at 4 °C in an airtight, light-protected container to maintain stability and prevent degradation prior to analysis. Chemical characterization was performed using an ISQ7610 Single Quadrupole GC–MS system (Thermo Scientific, Waltham, MA, USA) equipped with a 30 m fused silica capillary column (0.251 mm diameter, 0.1 mm film thickness). The analysis employed electron ionization at 70 eV, with helium serving as the carrier gas at a 1 mL/min flow rate. Component identification was achieved by comparing retention times and mass spectral data against NIST and WILLY library databases, with positive identification requiring match scores exceeding 80–90%. Relative peak areas were used to determine the proportions of chemical constituents [8].

Four experimental diets were formulated to be isonitrogenous (31.24 ± 0.17% crude protein) and isolipidic (8.22 ± 0.19% crude lipids). The experimental treatments consisted of the control (basal diet without supplementation), liposomal vitamin C (200 mg/kg) [31], CoQ10 (60 mg/kg) [30], and BV (4 mg/kg) [8]. The basal diet was formulated to meet the nutritional requirements of Nile tilapia [32], using protein sources, lipid sources, and carbohydrates, supplemented with dicalcium phosphate and a vitamin–mineral premix (Table 1). The diets were processed into 3 mm pellets, air-dried, and stored at 4 °C. The proximate composition analysis followed standard methods [33], with moisture determined by oven-drying (105 °C), ash by incineration (550 °C), crude lipids by Soxhlet extraction, crude protein by the Kjeldahl method, and crude fiber by sequential acid–alkali digestion followed by filtration and ashing [34]. Fish were fed to apparent satiation three times daily (07:00, 13:00, and 19:00) throughout the 60-day feeding trial.

### 2.4. Sampling and Performance Metrics

Following a 24 h fasting period, individual fish from each tank were weighed (body weight, BW), measured for total length (L), and counted. Growth performance and feed utilization were evaluated using the following equations:
Weight Gain (WG, g) = Final BW − Initial BW
$$WeightGainRate\;(WGR,\%)=\frac{FinalBW-InitialBW}{InitialBW}\times 100$$
Average Daily Gain (ADG, g) = WG ÷ Time (days)
Specific Growth Rate (SGR, %/day) = [(ln Final BW − ln Initial BW) ÷ Time (days)] × 100
Survival Rate (SR, %) = (Final Number ÷ Initial Number) × 100
Fulton’s Condition Factor (K) = (BW ÷ L^3^) × 100
Feed Conversion Ratio (FCR) = Feed Intake ÷ Weight Gain

For physiological analyses, nine fish per tank (27 per treatment) were randomly selected and anesthetized using MS-222 (100 mg/L). Blood samples were collected from the caudal vessels using non-heparinized syringes, centrifuged (3000× *g*, 5 min, 4 °C), and the resulting serum was stored at −80 °C until analysis. The following organ indices were calculated:Hepatosomatic Index (HSI, %) = (Liver Weight ÷ Body Weight) × 100Viscerosomatic Index (VSI, %) = (Viscera Weight ÷ Body Weight) × 100Intestinosomatic Index (ISI, %) = (Intestine Weight ÷ Body Weight) × 100

Gastrointestinal tissues were excised, rinsed with ice-cold PBS, homogenized, and centrifuged (6800× *g*, 5 min, 4 °C). The supernatants were stored at 4 °C for subsequent enzyme analysis. Liver tissues were homogenized in 0.86% NaCl solution, centrifuged (13,600× *g*, 10 min, 4 °C), and the supernatants were preserved for antioxidant analysis.

### 2.5. Digestive Enzyme and Microbial Determinations

Intestinal enzyme activities were measured at 37 °C. Protease activity was determined using casein as the substrate [35]. Lipase activity was measured using an olive oil emulsion in phosphate buffer, with absorbance at 714 nm. Amylase activity was assessed using a starch solution, with absorbance measured at 540 nm after reaction with a color reagent [36]. For microbial analysis, intestines from seven fish per replicate (n = 3) were pooled, rinsed with 0.9% saline, homogenized, diluted, and plated on selective media. Total bacteria were counted on plate count agar (30 °C, 24–72 h), *E. coli* on fecal coliform agar (30 °C, 24–48 h), *Vibrio* sp. on TCBS agar (37 °C, 18 h), and lactic acid bacteria on MRS medium (37 °C, anerobic, 72 h) [37]. Bacterial identification was based on morphological and biochemical tests as per Bergey’s Manual of Determinative Bacteriology.

### 2.6. Hepatointestinal Histology Evaluation

Liver and intestinal tissue samples were fixed in 10% neutral buffered formalin (pH 7.4) for 48 h. The fixed tissues were dehydrated through an ascending ethanol gradient (70–100%), cleared with xylene, and embedded in paraffin wax. To ensure uniformity, paraffin-embedded tissue blocks were properly oriented and evenly trimmed before sectioning. Tissue sections (5 μm thickness) were prepared using a calibrated rotary microtome (Leica RM2035, Wetzlar, Germany) with a well-maintained blade to ensure consistency. The sections were then stained with hematoxylin and eosin following standard protocols [38]. Stained sections were examined under a light microscope (Leica DM500) at 100× magnification, and digital images were captured using a Leica EC3 camera.

### 2.7. Innate Immunity and Antioxidant Determinations

Serum immune parameters were measured by assessing lysozyme activity with *Micrococcus lysodeikticus* [39], bactericidal activity by incubating serum with *E. coli* and measuring growth inhibition at 570 nm [8], and neutrophil oxidative burst using the NBT reduction assay [40]. Liver antioxidant enzyme activities (superoxide dismutase, SOD; catalase, CAT; glutathione peroxidase, GPx) and lipid peroxidation (malondialdehyde, MDA) were measured using commercial kits from Nanjing Jiancheng Institute of Bioengineering (Nanjing, China): SOD (WST-1 reagent, No#A001-3-2), CAT (ammonium molybdate reagent, No#A007-1-1), GPx (No#A005-1-2), and MDA (TBA reagent, No#A003-1-1), with absorbance measured at 550 nm, 280 nm, 412 nm, and 532 nm, respectively. Protein content was determined using the Bradford method.

### 2.8. Statistical Analysis

Prior to analysis, all data were tested for normality using a Shapiro–Wilk test and homogeneity of variance using a Levene’s test (α = 0.05). Treatment effects were analyzed using a one-way ANOVA followed by a Duncan’s multiple range test (*p* < 0.05). The results are presented as the mean ± standard error (n = 3). For microbial data, regression analysis and Pearson’s correlation coefficients were calculated to examine relationships between nutraceutical supplementation and bacterial populations. Data visualization, including scatter plots and heatmaps, was performed using Python (version 3.9) and R (version 4.3.1).

## 3. Results

### 3.1. Bee Venom Composition

The chemical composition of the bee venom extract revealed a wide variety of structures, including complex polycyclic systems, porphyrins, steroids, and esters. The most abundant compound, 9-Octadecen-1-ol (RT = 40.96), constituted over a quarter of the total peak area (28.35%), followed by Aralionine (RT = 49.94, 5.16%). Noteworthy compounds included Dotriacontane (RT = 44.56, 1.55%), a long-chain hydrocarbon, and 1,2-Benzenedicarboxylic acid, di-isooctyl ester (RT = 45.34, 4.45%), commonly used as a plasticizer. Additionally, the analysis identified bioactive substances such as Astaxanthin (RT = 48.23, 1.6%) and structural molecules like Flavone 4′-OH, 5-OH, 7-di-O-glucoside (RT = 47.34, 3.43%) (Figure 1 and Table 2).

### 3.2. Performance Metrics

The dietary supplementation of nutraceuticals significantly influenced the growth performance of Nile tilapia over the 60-day feeding trial (Table 3). Fish fed diets supplemented with CoQ10 and bee venom exhibited significantly enhanced growth parameters compared to both the control and vitamin C groups, as evidenced by final body weight, weight gain rate (WGR, %), and specific growth rate (SGR). The vitamin C-supplemented group demonstrated intermediate performance, showing significant improvements over the control group but remaining lower than the CoQ10 and bee venom treatments (*p* < 0.05). The feed conversion ratio (FCR) showed a similar trend, with all supplemented groups performing better than the control and bee venom supplementation resulting in the most efficient feed utilization. Notably, survival rates remained high across all treatments. The somatic indices (HSI, ISI, and VSI) and Fulton’s condition factor remained statistically comparable among all experimental groups.

### 3.3. Gastrointestinal Digestive Enzymes

The analysis of digestive enzyme activities revealed significant differences among the experimental groups following the 60-day feeding trial (Table 4). Bee venom supplementation resulted in the highest amylase activity compared to all other treatments, while both vitamin C and CoQ10 groups showed moderately enhanced amylase levels relative to the control (*p* < 0.05). For lipase and protease activities, both CoQ10 and bee venom supplementations demonstrated similarly elevated levels, significantly higher than the vitamin C group (*p* < 0.05), which in turn showed improved enzyme activities compared to the control group.

### 3.4. Gut Microbiota

The analysis of microbial populations across different nutraceutical treatments revealed significant patterns in bacterial community structure and treatment effectiveness (Table 5). All supplemented groups exhibited a significant reduction in total yeast and mold count (TYMC) and total bacterial count (TBC) compared to the control, with bee venom supplementation showing the most pronounced effect. Conversely, acid-fermentative bacteria populations were significantly enhanced in all supplemented groups compared to the control (*p* < 0.05). Notably, potentially pathogenic bacteria, including *Vibrio* sp., *Escherichia coli*, *Aeromonas* sp., *Salmonella* sp., *Shigella* sp., *Staphylococcus* sp., and *Streptococcus* sp., were substantially reduced in all supplemented groups, with bee venom demonstrating the strongest inhibitory effect, followed by the CoQ10 and vitamin C treatments.

A principal component analysis (PCA) demonstrated that 91.7% of the total variance could be explained by the first two principal components (Figure 2), with PC1 (pathogenic load axis) accounting for 73.2% and PC2 (beneficial bacteria axis) explaining 18.5% of the variation. Each nutraceutical demonstrated distinct antimicrobial profiles, with bee venom showing the most potent effects, followed by CoQ10 and vitamin C. The total bacterial count (TBC) exhibited treatment-specific reductions (R^2^ = 0.897, *p* < 0.001), decreasing from 52.33 ± 3.06 CFU/g in the control to 31.67 ± 3.06 CFU/g (bee venom), 38.33 ± 3.06 CFU/g (CoQ10), and 42.67 ± 3.06 CFU/g (vitamin C). Similar patterns were observed in the total yeast and mold count (TYMC) (R^2^ = 0.825, *p* < 0.001), with reductions from 4.33 ± 0.58 CFU/g in the control to 0.67 ± 0.58 CFU/g (bee venom), 1.33 ± 0.58 CFU/g (CoQ10), and 2.67 ± 0.58 CFU/g (vitamin C). The beneficial bacterial populations, particularly acid-fermentative bacteria, showed differential responses to treatments (R^2^ = 0.648, *p* < 0.01). The highest increase was observed with bee venom (15.33 ± 1.53 CFU/g), followed by CoQ10 (13.67 ± 1.53 CFU/g) and vitamin C (11.33 ± 1.53 CFU/g), compared to the control (9.33 ± 1.53 CFU/g). Pathogenic bacteria demonstrated treatment-specific reductions, with bee venom showing the strongest effects (76.5–81.0% reduction), followed by CoQ10 (55.3–62.1% reduction) and vitamin C (32.1–45.6% reduction) across *Vibrio* sp. (R^2^ = 0.892, *p* < 0.001), *E. coli* (R^2^ = 0.776, *p* < 0.001), and *Salmonella* sp. (R^2^ = 0.847, *p* < 0.001).

The chord diagram analysis (Figure 3) revealed distinct interaction patterns for each treatment. While bee venom demonstrated the strongest antimicrobial effects against pathogenic microorganisms, CoQ10 showed particular effectiveness against *Staphylococcus* sp. and *Aeromonas* sp., and vitamin C exhibited notable impacts on *Streptococcus* sp.

The heatmap of the correlation matrix for microbial counts reveals significant relationships among various microbial species in Nile tilapia (Figure 4). Total bacterial count (TBC) shows strong positive correlations with several species, including *Vibrio* sp. (r = 0.95), *Streptococcus* sp. (r = 0.96), and *Salmonella* sp. (r = 0.94), indicating that as TBC increases, these species also tend to increase. Conversely, acid-fermentative bacteria exhibit strong negative correlations with *Salmonella* sp. (r = −0.91) and *Shigella* sp. (r = −0.91), suggesting an inverse relationship. The correlation between total yeast and mold count (TYMC) and *Staphylococcus* sp. (r = 0.93) is also notably high. These correlations indicate intricate interactions among microbial populations under different treatment conditions, reflecting the complex dynamics of the gut microbiota in response to dietary interventions.

The regression analysis (Figure 5) provided further insights into treatment-specific effects. CoQ10 demonstrated strong relationships with TBC (R^2^ = 0.75) and pathogenic bacteria reduction (R^2^ = 0.70–0.72), while vitamin C showed moderate to strong associations with TBC (R^2^ = 0.68) and variable effects on pathogenic bacteria (R^2^ = 0.62–0.70). Both supplements maintained positive associations with beneficial bacteria populations (R^2^ = 0.65–0.70), though less pronounced than bee venom (R^2^ = 0.78). These comprehensive findings demonstrate the differential impacts of each nutraceutical treatment, with bee venom showing the most pronounced effects, followed by CoQ10 and vitamin C, in modulating the intestinal microbiota of Nile tilapia while maintaining beneficial bacterial communities.

### 3.5. Histological Assessment

The intestinal microscopic structure of Nile tilapia after 60 days of dietary treatment, as shown in Figure 6, revealed a normal, intact intestinal morphology across all groups. The intestinal mucosa, villi, and intestinal wall were well-preserved in all groups. Noteworthy, all supplemented groups exhibited an increase in villous areas and a higher number of goblet cells compared to the control group. Figure 7 illustrates the hepatic tissue structure of Nile tilapia after 60 days of dietary treatment. The hepatic tissue showed a characteristic spongy morphology with well-preserved hepatocytes and vesicular nuclei positioned centrally within the cells. Central veins were clearly defined, with leukocyte accumulation observed in the surrounding blood vessels. All supplemented groups demonstrated significant improvements in hepatic parenchyma, including pronounced glycogen accumulation, compared to the control group (*p* < 0.05).

### 3.6. Immune Function

The evaluation of immune function markers revealed the significant immunomodulatory effects of dietary supplementation in Nile tilapia (Figure 8). The lysozyme activity showed a stepwise increase across treatments, with bee venom supplementation eliciting the highest response, followed by CoQ10 and vitamin C, all significantly higher than the control group (*p* < 0.05). Similarly, serum bactericidal activity demonstrated the same hierarchical pattern, with bee venom supplementation inducing the strongest response, while CoQ10 and vitamin C treatments showed intermediate levels of enhancement compared to the control. The burst activity measured by nitroblue tetrazolium (NBT%) also exhibited significant improvements in all supplemented groups compared to the control (*p* < 0.05), with bee venom supplementation showing the highest activity, while vitamin C and CoQ10 treatments demonstrated similar moderate enhancements.

### 3.7. Antioxidant Responses

The hepatic antioxidant defense system and lipid peroxidation markers showed significant responses to dietary supplementation (Figure 9). Superoxide dismutase (SOD) activity was significantly enhanced in all supplemented groups (*p* < 0.05), with vitamin C and CoQ10 showing the highest levels, followed by the bee venom treatment, all surpassing the control group. Similarly, catalase (CAT) activity was markedly elevated in the vitamin C- and CoQ10-supplemented groups, while bee venom supplementation showed a moderate enhancement compared to the control. Glutathione peroxidase (GPx) activity was significantly increased in all supplemented groups compared to the control, with comparable levels among the three treatments (*p* < 0.05). Conversely, malondialdehyde (MDA) levels were significantly reduced in all supplemented groups compared to the control (*p* < 0.05), with similar levels of reduction observed across all treatments.

## 4. Discussion

Natural feed additives are increasingly favored over industrial additives and antibiotics in aquaculture due to their ability to enhance growth, feed efficiency, gut health, immunity, and disease resistance in fish [41]. In recent years, numerous natural additives have been investigated for their effects on fish health and performance [6,42]. Specifically, VC, CoQ10, and BV have shown significant benefits, including immune support, antioxidant effects, and antimicrobial properties [14,18,26]. However, the challenge for farmers and feed manufacturers lies in selecting the most effective additive for specific conditions [43]. The present study addresses this issue by comparing the effects of these additives on Nile tilapia performance and health, offering practical insights for aquafeed optimization.

Our study demonstrated that dietary supplementation with CoQ10 and BV significantly enhanced growth performance and feed efficiency in Nile tilapia compared to VC and the control group, with BV exhibiting the most pronounced effects. Despite these improvements, survival rate remained consistently high across all groups, and somatic indices showed no significant variations, highlighting the safety and tolerability of these supplements. These findings align with previous research where CoQ10 supplementation led to notable enhancements in the final body weight, weight gain, specific growth rate, and feed conversion ratio of mullet, with the highest performance recorded at dietary levels of 40 mg/kg and 60 mg/kg [20]. Similarly, earlier studies have reported improved growth performance in fish species (Nile tilapia, *O. niloticus*, and European seabass, *Dicentrarchus labrax* L.) fed CoQ10-enriched diets [9,30,44]. These improvements can be attributed to CoQ10’s role in enhancing mitochondrial bioenergetics by supporting the electron transport chain and ATP synthesis while mitigating oxidative stress, thereby optimizing energy metabolism and promoting growth [45].

Regarding BV supplementation, our results showed improved growth performance and confirmed its safety in fish diets, with even the lowest dose being effective [8,26]. The growth-promoting effects of BV are supported by research in other species, such as broilers [46] and rabbits [47], where BV supplementation enhanced body weight gain and survivability. These effects are largely attributed to the bioactive components of BV, particularly melittin, which exhibits antimicrobial and immunomodulatory properties. Melittin’s ability to maintain immune system functionality while creating a favorable physiological environment may explain the improved growth performance observed in this study [48]. Similarly, VC supplementation also positively affected growth performance and feed utilization. Previous studies have shown similar benefits in various fish species, including largemouth bass (*Micropterus salmoides*) [49], loach (*Misgurnus anguillicaudatus* Cantor) [50], and farmed tilapia (*O. niloticus*) [51]. VC’s beneficial effects can be attributed to its antioxidant properties, its critical role in collagen synthesis, and its capacity to enhance immune responses, all of which collectively contribute to improved growth and feed efficiency [14].

Gut microbiota and digestive enzymes in fish play crucial roles in nutrient breakdown, absorption, and feed efficiency [52]. The microbiota supports immune function and growth [53], while digestive enzymes improve feed digestibility [54]. In our study, BV supplementation resulted in the highest amylase activity, while CoQ10 and VC showed moderate improvements. Both BV and CoQ10 elevated lipase and protease activities, outperforming VC. All supplemented groups had lower yeast, mold, and bacterial counts, with BV having the strongest effect. Additionally, acid-fermentative bacteria increased, and pathogenic bacteria were significantly reduced, with BV showing the most pronounced inhibition. These findings align with El Basuini et al. [8], who observed that lipase and protease activity reached maximum levels across all concentrations of BV in mullet, with no significant differences between BV levels. However, amylase activity remained unchanged across all groups, and El Basuini et al. [26] found no alterations in amylase, protease, or lipase activities in Nile tilapia following BV injection. For CoQ10, El Basuini et al. [20] reported heightened amylase, lipase, and protease activities in fish fed CoQ10-enriched diets compared to the controls. Similarly, El Basuini et al. [30] found increased digestive enzyme activities in Nile tilapia when CoQ10 was supplemented at 20–40 mg/kg.

The growth-promoting effects of CoQ10 may stem from its influence on microflora diversity, the re-synthesis of vitamin E, anti-inflammatory actions, and digestive enzyme activity [55,56], as well as its impact on lipid, protein, and carbohydrate metabolism through the electron transport chain [57]. CoQ10’s effects on hormones such as glucagon, insulin, and cortisone may also contribute to improved fish performance [58]. As for VC, similar improvements in enzyme activity and intestinal microflora were reported in other studies [59,60,61], highlighting the vitamin’s role as a cofactor in key metabolic processes.

A histological examination of the intestine and liver is essential for assessing the structural integrity and functional health of fish organs [62]. In the present study, after 60 days of dietary treatment, Nile tilapia showed a well-preserved intestinal structure across all groups, with the supplemented groups exhibiting enhanced villous areas and an increased density of goblet cells compared to the control. These findings align with previous studies, such as El Basuini et al. [8], which demonstrated intact intestinal structures in grey mullet across all experimental groups, indicating healthy gastrointestinal function crucial for nutrient absorption [63]. The observed enhancement of intestinal villi following BV supplementation suggests beneficial effects on intestinal health and feed utilization, which are vital for optimal fish growth [26,64]. Additionally, increasing levels of CoQ10 supplementation corresponded with improvements in villous height and branching, with higher doses (60 and 80 mg/kg) showing a prominent lymphoepithelium near the base of the villi [20]. Similarly, El Basuini et al. [44] observed the positive effects of CoQ10 on the intestinal histology of European seabass. The current findings regarding VC supplementation are consistent with earlier studies showing histological improvements in the intestines of fish fed VC [65,66].

In the liver, all supplemented groups showed well-organized hepatic architecture with well-preserved hepatocytes and central veins, characterized by improved glycogen accumulation and hepatic parenchyma. These histological changes suggest enhanced metabolism and fish health, as the liver plays a crucial role in energy storage, detoxification, and growth-related protein synthesis [67]. BV supplementation resulted in noticeable changes in liver histopathology, including improved hepatic architecture and glycogen deposition. However, hepatocyte vacuolation was observed at higher BV doses, indicating potential adverse effects [8,26]. CoQ10 supplementation induced a spongy liver appearance with intact hepatocytes around the central veins, and progressive glycogen buildup up to 60 mg/kg. However, at the highest dose of 80 mg/kg, vascular dilation and the mild vacuolation of hepatocytes were observed [19,56]. These results align with earlier studies on African catfish fed 40 mg/kg CoQ10, which showed the preservation of normal liver histology [55]. The observed improvements in liver histology with VC supplementation corroborate previous findings [66,68].

The immune response is critical for fish health, and nutrient supplementation can enhance innate immune parameters, including lysozyme activity, bactericidal activity, and NBT burst activity [69,70,71]. In this study, dietary supplementation with BV, CoQ10, and VC significantly elevated these immune parameters compared to the control, with BV exhibiting the most pronounced effects. These results align with previous research on mullet, where fish fed a BV-supplemented diet demonstrated the highest levels of lysozyme, bactericidal, and NBT burst activity [8]. BV’s immunomodulatory effects are attributed to its bioactive components, which activate and enhance immune responses [64]. Similar benefits have been observed in other species, supporting the potential therapeutic use of BV for immunotherapy [23,72]. CoQ10 supplementation at 60 mg/kg diet has also been shown to significantly enhance immune parameters like lysozyme levels, bactericidal activity, and NBT, as demonstrated in mullet [20] and other studies on various fish species [30,55,56]. VC supplementation at 200 mg/kg diet has likewise been reported to improve innate immunity in fish, including the enhancement of lysozyme activity, complement activity, and respiratory burst [14,73].

In terms of antioxidant defense, enzymes such as SOD, CAT, and GPx play a critical role in neutralizing free radicals and maintaining the cellular redox balance, thereby protecting against oxidative damage [SOD, [74]; CAT, [75], and GPx, [76]]. MDA levels, a marker of lipid peroxidation, provide insights into oxidative stress [MDA, [77]]. In this study, all supplemented groups exhibited significantly enhanced SOD, CAT, and GPx activities, with VC at 200 mg/kg and CoQ10 at 60 mg/kg showing the highest levels. MDA levels were significantly reduced in all supplemented groups, with comparable reductions across treatments. These findings are consistent with previous research, including studies on BV supplementation at 4 mg/kg in mullet, which showed increased antioxidant enzyme activities and reduced MDA levels compared to the controls [8]. The antioxidant effects of BV have been widely reported in various species, including rabbits and poultry, where BV enhanced antioxidant capacity and protected against oxidative damage [47,73]. Furthermore, CoQ10 is known for its potent antioxidant properties, and its supplementation has been shown to elevate SOD, CAT, and GPx activities while reducing MDA in fish and other animals [9,78]. Previous studies have also demonstrated that dietary CoQ10 supplementation in fish, such as Nile tilapia and African catfish, enhances antioxidant markers and mitigates oxidative stress [55,56,79]. Similarly, VC supplementation has been shown to enhance antioxidant enzyme activities in various fish species [14,80,81].

## 5. Conclusions

Dietary supplementation with bee venom (4 mg/kg diet), coenzyme Q10 (60 mg/kg diet), and vitamin C (200 mg/kg diet) significantly improved growth performance, immune function, antioxidant capacity, and intestinal health in Nile tilapia over a 60-day feeding trial. BV and CoQ10 showed superior effects on growth, digestive enzyme activity, and gut microbiota balance, with BV exhibiting the strongest antimicrobial properties. Histological analysis confirmed enhanced intestinal and hepatic health in all supplemented groups. Immunological and antioxidant responses were significantly elevated, with BV providing the strongest immune boost, while CoQ10 and VC enhanced antioxidant defense. Based on these findings, BV (4 mg/kg) is recommended for immune enhancement, CoQ10 (60 mg/kg) for growth and antioxidant protection, and VC (200 mg/kg) for overall health maintenance. Further studies should assess the long-term impacts and economic feasibility in commercial tilapia production.

## Figures and Tables

**Figure 1 biology-14-00309-f001:**
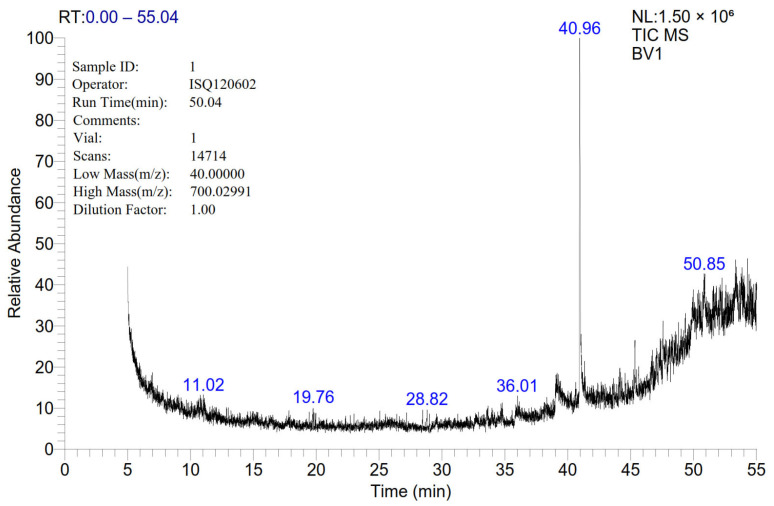
Gas chromatography–mass spectrometry (GC–MS) analysis of the bee venom (BV) extract in full-scan mode, showing the representative chromatogram of the dimethyl sulfoxide (DMSO)-based extract.

**Figure 2 biology-14-00309-f002:**
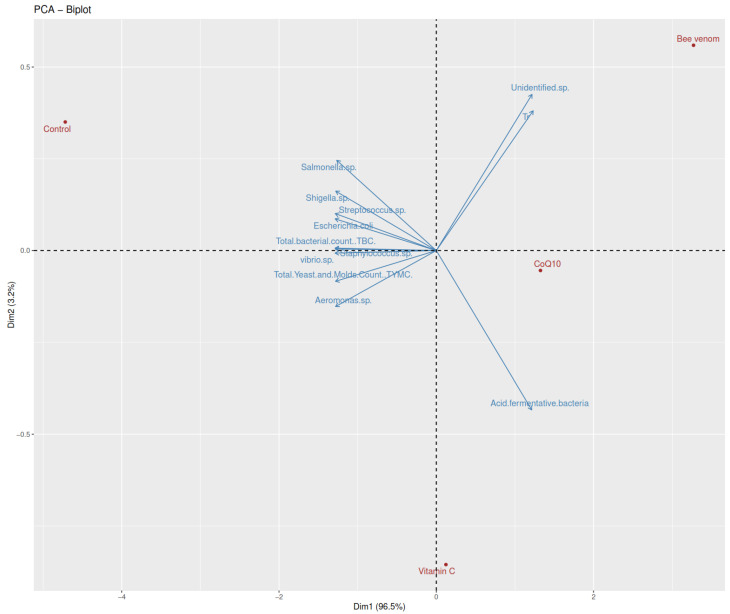
Principal component analysis (PCA) of microbial community composition in Nile tilapia (*Oreochromis niloticus*) intestines following the 60-day dietary supplementation.

**Figure 3 biology-14-00309-f003:**
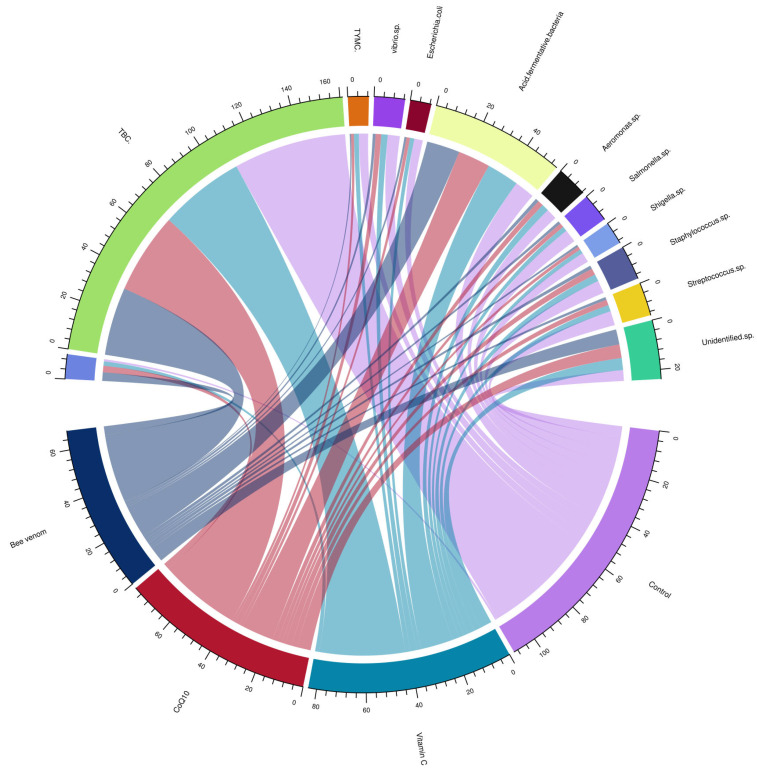
Chord diagram of the interaction patterns between nutraceutical treatments and microbial populations in Nile tilapia (*Oreochromis niloticus*) intestines. TBC: total bacterial count; TYMC: total yeast and mold count.

**Figure 4 biology-14-00309-f004:**
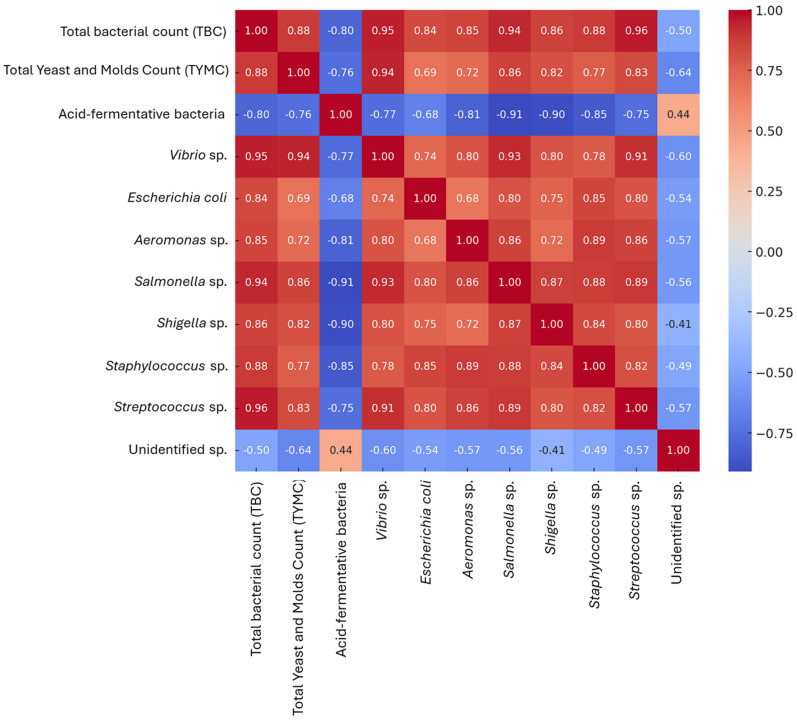
Heatmap correlation analysis of gut microbiota counts in Nile tilapia (*Oreochromis niloticus*) after a 60-day feeding trial.

**Figure 5 biology-14-00309-f005:**
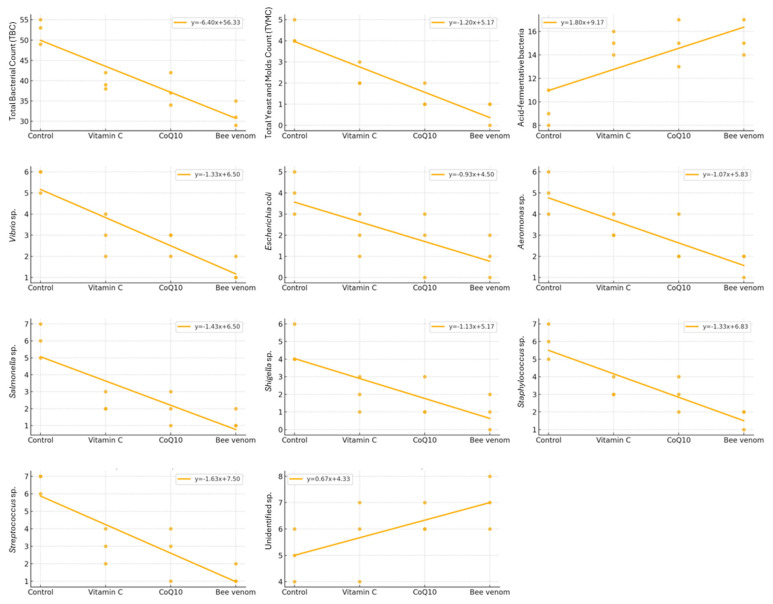
Regression analysis between dietary supplements and the intestinal microbial counts of Nile tilapia (*Oreochromis niloticus*).

**Figure 6 biology-14-00309-f006:**
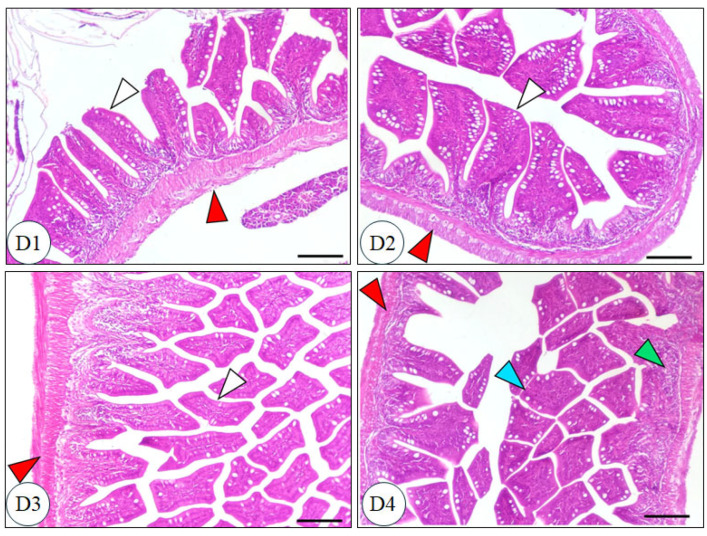
Intestinal microscopic structure of Nile tilapia (*Oreochromis niloticus*) after 60 days of dietary treatment, stained with hematoxylin and eosin (H&E) at a 100 µm scale. D1 = basal diet without supplements (control); D2 = basal diet + 200 mg liposomal vitamin C (LVC, ASIN: B00JFF48I6); D3 = basal diet + 60 mg CoQ10; D4 = basal diet + 4 mg bee venom. Green arrowhead: intestinal mucosa; Red arrowhead: intestinal wall; Blue arrowhead: villi; White arrowhead: villous area.

**Figure 7 biology-14-00309-f007:**
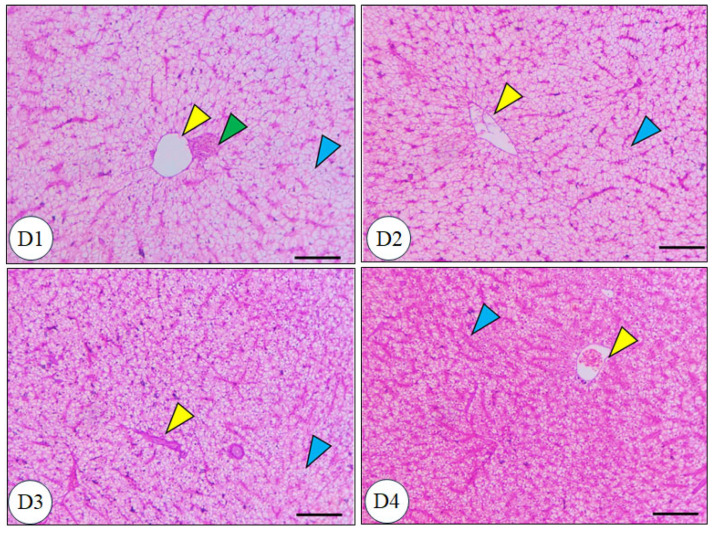
Microscopic structure of Nile tilapia (*Oreochromis niloticus*) hepatic tissue following 60 days of dietary treatment (H&E staining, 100 µm scale). D1 = basal diet without supplements (control); D2 = basal diet + 200 mg liposomal vitamin C (LVC, ASIN: B00JFF48I6); D3 = basal diet + 60 mg CoQ10; D4 = basal diet + 4 mg bee venom. Yellow arrowhead: central veins; Green arrowhead: leukocytes; Blue arrowhead: hepatocytes containing vesicular nuclei.

**Figure 8 biology-14-00309-f008:**
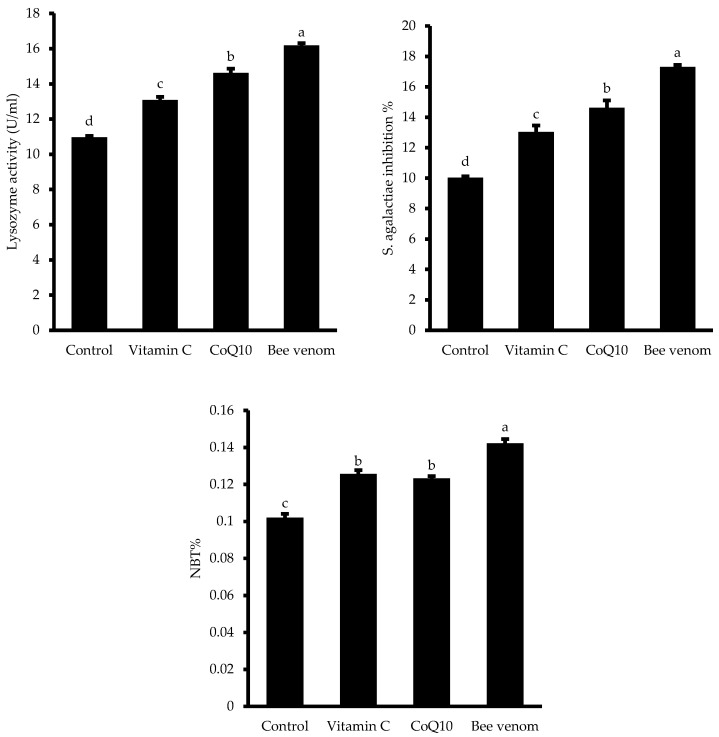
Immune function markers in Nile tilapia (*Oreochromis niloticus*) after a 60-day dietary trial. Bars labelled with different letters (a, b, c, d) represent statistically significant variations (*p* < 0.05).

**Figure 9 biology-14-00309-f009:**
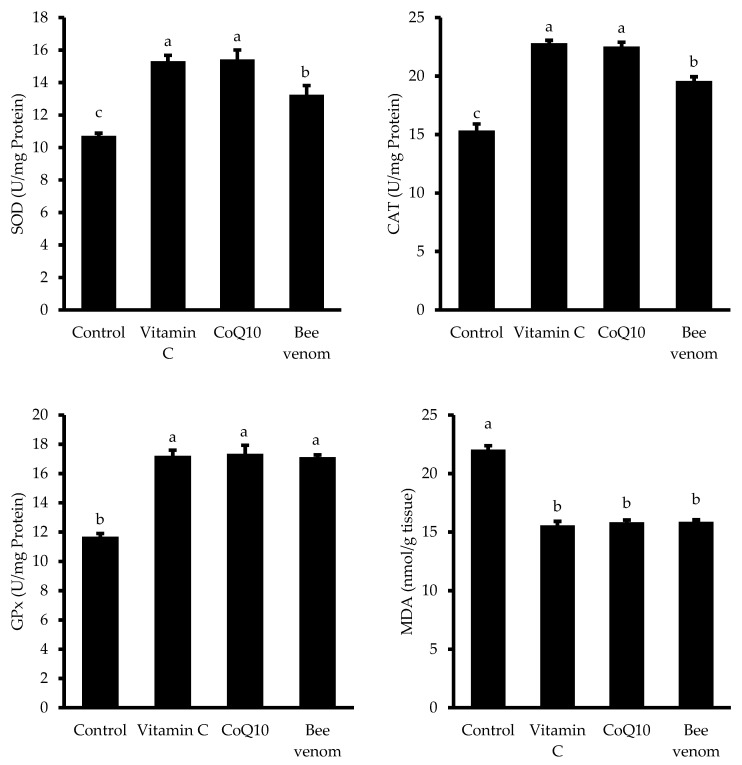
Hepatic antioxidant defense systems and lipid peroxidation markers in Nile tilapia (*Oreochromis niloticus*) following a 60-day dietary trial. SOD (superoxide dismutase), CAT (catalase), GPx (glutathione peroxidase), and MDA (malondialdehyde). Bars labelled with different letters (a, b, c) represent statistically significant variations (*p* < 0.05).

**Table 1 biology-14-00309-t001:** Ingredients and nutrient profile of the basal diet.

Ingredients	%
Soybean meal (44% CP)	35
Yellow corn	20
Fish meal (65% CP)	15
Wheat bran	7
Wheat flour	6
Rice bran	5
Gluten	5
Fish oil	3
Soybean oil	2
Dicalcium phosphate	1
Vitamin and mineral premix ^1^	1
Total	100
Nutrient profile	
Crude protein (%)	31.24 ± 0.17
Crude lipids (%)	8.22 ± 0.19
Fiber (%)	4.15 ± 0.12
Ash (%)	7.3 ± 0.05
Gross energy (MJ/Kg) ^2^	18.16 ± 0.12

^1^ Vitamin and mineral mixture detailed by El Basuini et al. [9]. ^2^ The gross energy was estimated using the values of 23.6 kJ/g for protein, 39.5 kJ/g for lipids, and 17.2 kJ/g for carbohydrates.

**Table 2 biology-14-00309-t002:** GC–MS chemical composition analysis of the bioactive compounds in the *Apis mellifera* venom extract.

RT (min)	Compound	Molecular Formula	Mass (*m*/*z*)	Area (%)
6.88	5-HCyclopropa[3,4]benz[1,2-e]azulen-5-one,4,9,9-atris(acetyloxy)3[(acetyloxy)methyl]1,1a,1b,4,4a,7a,7b,8,9,9a- decahydro-4a,7-bdihydroxy1,1,6,8-tetramethyl	C_28_H_36_O11	549	0.78
7.82	2,2Bis[4[(4,6dichloro1,3,5triazin2yl)oxy]phenyl]1,1,1,3,3,3hexafluoropropane	C_21_H_8_C_l4_F_6_N_6_O_2_	632	0.82
9.26	Hycanthone	C_20_H_24_N_2_O_2_S	356	0.81
10.79	4,5,6,7Tetrakis(pchlorophenoxy)1,2diiminoisoindoline	C_32_H_19_C_l4_N_3_O_4_	10.79	1.11
11.11	Copper tetraphenylporphyrin	C_44_H_28_CuN_4_	676	0.9
11.79	3,4,10,11tetrakis(Dimethylamino)7,14bis(trifluoromethyl)7,14epoxydinaphtho[1,8ab:1′,8′ef]cyclooctane	C_32_H_32_F_6_N_4_O	603	0.78
11.95	Decanoicacid,1,1a,1b,4,4a,5,7a,7b,8,9decahydro4a,7bdihydroxy1,1,6,8tetramethyl5oxo3[[(1oxodecyl)oxy]methyl]9aHc yclopropa[3,4]benz[1,2e]azulene9,9adiylester	C_50_H_82_O_9_	827	1.05
17.83	2-Myristynoyl pantetheine	C_25_H_44_N_2_O_5_S	484	1.2
19.76	2,4bis(áchloroethyl)6,7bis[ámethoxycarbonylethyl]8formyl1,3,5trimethylporphyrin	C_36_H_38_C_l2_N_4_O_5_	677	1.03
32.66	Tetraneurin-A-diol	C_15_H_20_O_5_	280	0.83
33.55	Tristrimethylsilyl ether derivative of 1,25-dihydroxyvitamin D2	C_37_H_68_O_3_Si_3_	645	1.16
33.65	Pregn-4-ene-3,11,20trione,6,17,21-tris[(trimethylsilyl)oxy]-,3,20-bis(O-methyloxime)	C_32_H_58_N_2_O_6_Si_3_	651	0.91
35.58	16-Oxapentacyclo[13.2.2.0(1,13).0(2,10).0(5,9)]nonadecane	C_22_H_34_D_2_O_3_	346	0.74
35.88	Trans-2-phenyl-1,3-dioxolane-4-methyloctadec-9,12,15-trienoate	C_28_H_40_O_4_	440	0.88
38.28	2-Cyclohexyl-4a,7-dimethyl-3,4,4a,5,6,8a-hexahydro-2H-benzo[e][1,2]oxazine-3-carbonitrile	C_17_H_26_N_2_O	274	0.81
39.03	Butanoicacid,4-chloro,1,1a,1b,4,4a,5,7a,7b,8,9-decahydro-4a,7b-dihydroxy-3-(hydroxymethyl)-1,1,6,8-tetramethyl-5-oxo-9a-Hcyclopropa[3,4]benz[1,2-e]azulene9,9a-diylester	C_28_H_38_C_l2_O_8_	573	0.76
39.16	4-(-1-hydroxyethyl)-1,6,7-tris-(2-methoxycarbonylethyl)-2,3,5,8-tetramethylporphyrin	C_38_H_44_N_4_O_7_	668	1.23
40.96	9-Octadecen-1-ol,(Z)-(CAS)	C_18_H_36_O	268	28.35
42.38	6-C-Xylosyl-8-C-glucosylapigenin-permethylated derivative	C_33_H_36_O_17_	704	1.43
43.64	5á-Pregnan-20-one,3à,11á,17,21-tetrakis(trimethylsiloxy)-,O-methyloxime	C_34_H_69_NO_5_Si_4_	684	1.2
44.13	(22S)-21-Acetoxy-6à-,11ádihydroxy16à,17à-propylmethylenedioxypregna-1,4-diene-3,20-dione	C_27_H_36_O_8_	488	1.15
44.2	N,N′-Dicyclohexyl-1,7-dipyrrolidinylperylene-3,4:9,10-tetracarboxylicacid bisimide	C_44_H_44_N_4_O_4_	692	0.73
44.56	Dotriacontane (CAS)	C_32_H_66_	450	1.55
44.81	Isochiapin B	C_19_H_22_O_6_	346	1.01
45.34	Benzene, 2(1decyl1undecenyl)1,4dimethyl (CAS)	C_29_H_50_	399	4.45
45.65	(5,10,15,20-tetraphenyl[2-(2)H1]prophyrinato)zinx(II)	C_44_H_27_DN_4_Zn	677	1.14
46.64	3,5,9-Trioxa-5-phosphaheptacos-18-en-1-aminium,4-hydroxy-N,N,N-trimethyl-10-oxo-7-[(1-oxo-9-octadecenyl)oxy]-	C_44_H_84_NO_8_P	786	1.35
46.69	3-Hydroxy-1-(4{13-[4-(3-hydroxy-3-phenylacryloyl)phenyl]tridecyl}-phenyl)-3-phenylprop-2-en-1-one	C_43_H_48_O_4_	628	1.06
46.73	Pregn-4-ene-3,20-dione, 17,21-dihydroxy-,bis(Omethyloxime)	C_23_H_36_N_2_O_4_	404	0.88
47.09	Corynan-17-ol,18,19-didehydro-10-methoxy-,acetate (ester)	C_22_H_28_N_2_O_3_	368	2.19
47.34	Flavone 4′-oh, 5-oh, 7-di-o-glucoside	C_27_H_30_O_15_	594	3.43
47.58	4,25-Secoobscurinervan-21-deoxy-16-methoxy-22-methyl-,(22à)-(CAS)	C_23_H_32_N_2_O_2_	368	3.89
47.86	Fucoxanthin	C_42_H_58_O_6_	658	1.06
48.08	4H-Cyclopropa[5′,6′]benz[1′,2′:7,8]azuleno[5,6-b]oxiren-4-one,8,8abis(acetyloxy)-2a-[(acetyloxy)methy-l]	C_26_H_34_O_11_	522	3.07
48.23	Astaxanthin	C_40_H_52_O_4_	596	1.6
48.29	Benzene,2-(1-decyl-1-undecenyl)-1,4-dimethyl-(CAS)	C_29_H_50_	398	1.72
48.35	9-Octadecenoicacid,(2-phenyl-1,3-dioxolan-4-yl)methyl ester, cis-(CAS)	C_28_H_44_O_4_	444	1.77
48.53	(2-hydroxy-5,10,15,20-tetraphenylporphinato)zinc(II)	C_44_H_28_N_4_OZn	694	2.81
49.02	Ethyl iso-allocholate	C_26_H_44_O_5_	436	1.46
49.34	Tetraphenylporphyrinat odibromotitanium(IV)	C_44_H_28_Br_2_N_4_Ti	820	2.07
49.83	Stigmast-5-en-3-ol,(3á,24S)-(CAS)	C_29_H_50_O	414	1.19
49.94	Aralionine	C_34_H_38_N_4_O_5_	582	5.16

**Table 3 biology-14-00309-t003:** Performance metrics for Nile tilapia (*Oreochromis niloticus*) following a 60-day dietary trial.

Parameters	Control	Vitamin C	CoQ10	Bee Venom
Initial body weight, g	35.23 ± 0.26	35.24 ± 0.11	35.20 ± 0.23	35.12 ± 0.39
Final body weight, g	94.89 ± 2.31 ^c^	128.97 ± 3.70 ^b^	145.26 ± 2.53 ^a^	145.68 ± 2.26 ^a^
Weight gain rate (WGR, %)	169.43 ± 8.53 ^c^	266.02 ± 11.54 ^b^	312.67 ± 7.91 ^a^	315.05 ± 10.82 ^a^
Specific growth rate (SGR, %/day)	1.65 ± 0.05 ^c^	2.10 ± 0.03 ^b^	2.36 ± 0.03 ^a^	2.37 ± 0.04 ^a^
Feed conversion ratio (FCR)	1.96 ± 0.08 ^a^	1.56 ± 0.05 ^b^	1.42 ± 0.04 ^bc^	1.36 ± 0.01 ^c^
Survival rate (SR, %)	97.78 ± 2.22	98.89 ± 1.11	100.00 ± 0.00	100.00 ± 0.00
Hepatosomatic index (HSI, %)	2.10 ± 0.03	2.08 ± 0.12	2.14 ± 0.12	2.13 ± 0.13
Intestinosomatic index (ISI, %)	3.35 ± 0.12	3.46 ± 0.22	3.40 ± 0.29	3.41 ± 0.13
Viscerosomatic index (VSI, %)	6.66 ± 0.14	6.60 ± 0.25	6.53 ± 0.38	6.50 ± 0.23
Fulton’s condition factor (K factor)	2.04 ± 0.12	2.04 ± 0.07	2.09 ± 0.11	2.07 ± 0.01

Values within the same row are the mean ± S.E. ^a^, ^b^, and ^c^ superscripts denote significance at *p* < 0.05.

**Table 4 biology-14-00309-t004:** Enzyme profiles in Nile tilapia (*Oreochromis niloticus*) post the 60-day dietary experiment.

Enzyme Activity (U/mg)	Control	Vitamin C	CoQ10	Bee Venom
Amylase	11.35 ± 0.33 ^c^	13.76 ± 0.25 ^b^	14.45 ± 0.45 ^b^	18.08 ± 0.35 ^a^
Lipase	12.16 ± 0.17 ^c^	16.42 ± 0.34 ^b^	22.14 ± 0.17 ^a^	22.06 ± 0.53 ^a^
Protease	11.77 ± 0.26 ^c^	13.13 ± 0.10 ^b^	16.95 ± 0.25 ^a^	16.30 ± 0.36 ^a^

Values within the same row are the mean ± S.E. ^a^, ^b^, and ^c^ superscripts denote significance at *p* < 0.05.

**Table 5 biology-14-00309-t005:** Microbial composition of Nile tilapia (*Oreochromis niloticus*) intestines after the 60-day test diets.

Microbiota Count (Log CFU/g)	Control	Vitamin C	CoQ10	Bee Venom
Total yeast and mold count (TYMC)	4.33 ± 0.33 ^a^	2.33 ± 0.33 ^b^	1.33 ± 0.33 ^bc^	0.67 ± 0.33 ^c^
Total bacterial count (TBC)	52.33 ± 1.76 ^a^	39.67 ± 1.20 ^b^	37.67 ± 2.33 ^b^	31.67 ± 1.76 ^c^
Acid-fermentative bacteria	9.33 ± 0.88 ^b^	15.00 ± 0.58 ^a^	15.00 ± 1.15 ^a^	15.33 ± 0.88 ^a^
*Vibrio* sp.	5.67 ± 0.33 ^a^	3.00 ± 0.58 ^b^	2.67 ± 0.33 ^b^	1.33 ± 0.33 ^c^
*Escherichia coli*	4.00 ± 0.58 ^a^	2.00 ± 0.58 ^ab^	1.67 ± 0.88 ^b^	1.00 ± 0.58 ^b^
*Aeromonas* sp.	5.00 ± 0.58 ^a^	3.33 ± 0.33 ^b^	2.67 ± 0.67 ^b^	1.67 ± 0.33 ^b^
*Salmonella* sp.	6.00 ± 0.58 ^a^	2.33 ± 0.33 ^b^	2.00 ± 0.58 ^b^	1.33 ± 0.33 ^b^
*Shigella* sp.	4.67 ± 0.67 ^a^	2.00 ± 0.58 ^b^	1.67 ± 0.67 ^b^	1.00 ± 0.58 ^b^
*Staphylococcus* sp.	6.00 ± 0.58 ^a^	3.33 ± 0.33 ^b^	3.00 ± 0.58 ^bc^	1.67 ± 0.33 ^c^
*Streptococcus* sp.	6.67 ± 0.33 ^a^	3.00 ± 0.58 ^b^	2.67 ± 0.88 ^b^	1.33 ± 0.33 ^b^
Unidentified sp.	5.00 ± 0.58	5.67 ± 0.88	6.33 ± 0.33	7.00 ± 0.58

Values within the same row are the mean ± S.E. ^a^, ^b^, and ^c^ superscripts denote significance at *p* < 0.05.

## Data Availability

The original contributions presented in this study are included in the article, and further inquiries can be directed to the corresponding authors.

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
