# Peer review of "Effects of Liposomal Vitamin C, Coenzyme Q10, and Bee Venom Supplementation on Bacterial Communities and Performance in Nile Tilapia (Oreochromis niloticus)"

_biology, 2025, doi:10.3390/biology14030309_

Round 1
Reviewer 1 Report
Comments and Suggestions for Authors
This study was designed to evaluate the impacts of three nutraceuticals on physiological parameters of Nile tilapia (Oreochromis niloticus). It can provide valuable insights into their potential applications for enhancing the efficiency and sustainability of tilapia farming systems. However, several problems need to be improved and explained in this paper.
In the abstract and M&M, what is the basis for choosing the level of addition of the three additives?
Values of feed energy should be measured, not theoretical values.
Please mark p-value for significance in the presentation of results.
Figures 2 and 3 are not clear.
The whole conclusion is too long and needs to be streamlined
Line 61-63, relevant studies have been confirmed in tilapia, especially CoQ10, so what are the innovations and research significance of this paper?
Line 108-110, missing relevant references.
Line 138, supplement the instruments and types of water quality testing.
Line 166, describe the methods of the crude fibre.
Line 177, based on the growth results in Table 3, it is shown that WG should be modified to WGR and the formula should be re-corrected.
Line 427, add the English names of the fish species.
Line 428, ‘with comparable results in broiler’, such comparisons should be deleted, there seems to be no comparability between species.
Line 442, add the latin names.
Author Response
Response to Reviewer #1: -
This study was designed to evaluate the impacts of three nutraceuticals on physiological parameters of Nile tilapia (Oreochromis niloticus). It can provide valuable insights into their potential applications for enhancing the efficiency and sustainability of tilapia farming systems. However, several problems need to be improved and explained in this paper.
Response: -
- All authors thank the reviewer for his/her valuable comments and constructive reviews for improving our manuscript. Moreover, your insightful comments have significantly enhanced the overall quality of our manuscript. Please find the response to your comments “point-by-point” along with a revised version of our manuscript showing the changes required in light of your comments, newly added paragraphs, amended text, and updated references using “YELLOW HIGHLIGHTS”.
- We confirm that all your comments were taken carefully into consideration, and we believe that we replied to your comments to an appropriate degree.
Q1. In the abstract and M&M, what is the basis for choosing the level of addition of the three additives?
Task completed: ✔.
- Thank you for your insightful question. The additive levels were selected based on previous studies that have recommended these specific concentrations. The relevant references supporting our choice are as follows: (https://doi.org/10.1007/s10499-025-01882-4; https://doi.org/10.1016/j.fsi.2020.01.052; https://doi.org/10.1016/j.fsi.2024.109713). (kindly see the new lines 165-167).
Q2. Values of feed energy should be measured, not theoretical values.
Task completed: ✔.
- Thank you for your comment. The nutrient composition of the diet, including crude protein, crude lipids, fiber, and ash content, was analyzed to ensure accuracy (Lines 170-174). The gross energy (GE) values were estimated using established physiological fuel values (23.6 kJ/g for protein, 39.5 kJ/g for lipid, and 17.2 kJ/g for carbohydrates), which is a widely accepted approach in aquaculture nutrition studies. While direct measurement of GE through bomb calorimetry can provide more precise values, estimated GE remains a standard and reliable method in fish nutrition research.
Q3. Please mark p-value for significance in the presentation of results.
Task completed: ✔.
- Thank you for your valuable comments. We have incorporated the p-values as recommended. (kindly see the new lines 272, 288, 290, 302, 379, 398, 403, 413, 419, 420).
Q4. Figures 2 and 3 are not clear.
Task completed: ✔.
- Thank you for your feedback. We have improved the clarity and resolution of Figures 2 and 3 to enhance their readability. Please review the revised figures in the updated manuscript.
Q5. The whole conclusion is too long and needs to be streamlined.
Task completed: ✔.
- Thank you for your valuable suggestion. We have revised the conclusion to make it more concise while retaining the key findings and implications. (kindly see the new lines 554-565).
Q6. Line 61-63, relevant studies have been confirmed in tilapia, especially CoQ10, so what are the innovations and research significance of this paper?
Task completed: ✔.
- Thank you for your comment. While previous studies have investigated CoQ10 in tilapia, the novelty of this research lies in the comparative evaluation of liposomal vitamin C (VC), CoQ10, and bee venom (BV) on multiple key aspects of fish health and performance. This study is one of the first to directly compare these nutraceuticals in a single experimental framework, providing a comprehensive analysis of their effects on growth performance, immune responses, digestive enzyme activity, hepato-intestinal histology, and antioxidant defense mechanisms. Additionally, the use of liposomal VC represents an innovative approach, as liposomal formulations are designed to enhance bioavailability and stability, potentially leading to improved physiological responses. By identifying the most effective supplementation strategy, this study provides practical insights for optimizing fish nutrition and improving the sustainability of tilapia farming. Please check lines 116-122.
Q7. Line 108-110, missing relevant references.
Task completed: ✔.
- Thank you for your observation. We have now added the appropriate references to support the statements. (kindly see the new line 111).
Q8. Line 138, supplement the instruments and types of water quality testing.
Task completed: ✔.
- Thank you for your suggestion. We have specified the instruments and methods used for water quality testing. (kindly see the new lines 139-143).
Q9. Line 166, describe the methods of the crude fibre.
Task completed: ✔.
- Thank you for your comment. We have now included the methodology used for crude fiber determination. (kindly see the new lines 173-174).
Q10. Line 177, based on the growth results in Table 3, it is shown that WG should be modified to WGR and the formula should be re-corrected.
Task completed: ✔.
- Thank you for your observation. The necessary modification has been made, and WG has been updated to WGR with the corrected formula. (Kindly see the new lines 186, 269 and Table 3).
Q11. Line 427, add the English names of the fish species.
Task completed: ✔.
- Thank you for your suggestion. The English names of the fish species have been added accordingly. (Kindly see the new line 447).
Q12. Line 428, ‘with comparable results in broiler’, such comparisons should be deleted, there seems to be no comparability between species.
Task completed: ✔.
- Thank you for your observation. The comparison has been removed to maintain clarity and accuracy. (Kindly see the new line 448).
Q13. Line 442, add the latin names.
Task completed: ✔.
- Thank you for your suggestion. We have included the Latin names accordingly. (Kindly see the new lines 461-463).
Finally, all authors and I would like to thank you for your notice and valuable reviews, and we hope that we responded to all your comments to meet your expectations.
Reviewer 2 Report
Comments and Suggestions for Authors
- The author is suggested to italicize the scientific name and once use the full name after this use only O. niloticus throughout the manuscript. This makes the paper more worthy.
- Such type of mistakes makes the paper poor and less interesting for readers/reviewers please remove such kind of errors throughout the paper.
- Correct this error.
- Write the fiberglass dimension (what is the length, width, and height of these tanks) properly
- Correct this error
- Write number of fish/ aquarium that was added in each fiberglass tanks during trial
- Were these products tested for purity and potency?
- What specific parameters were used for the semiautomatic collection system (e.g., voltage, pulse duration)?
- How were the storage conditions for the purified bee venom maintained at 4°C?
- How were the nutritional requirements of Nile tilapia determined for the formulation of the basal diet
- Most of the time, fish were fed only twice a day. Why are you giving feed 3 times any specific reason? And when Siphoning was done daily to maintain water quality.
- Correct this sentence.
- For amylase and lipase activity write few lines about the procedure i.e solution preparation etc.
- How was the consistency of the 5 μm thick tissue sections ensured during preparation with the rotary microtome (Leica RM2035)?
- It is noticed that some values in this table do not have superscripts so add superscripts from SR to K
- Write the scientific name it will be more decent and make paper more presentable.
- Please also label the histology slides as written below.
- It is suggested to the author that many grammatical mistakes need to be corrected.

English quality must be improved.
Author Response
Response to Reviewer #2:
All authors thank the reviewer for his/her valuable comments and constructive reviews for improving our manuscript. Moreover, your insightful comments have significantly enhanced the overall quality of our manuscript. Please find the response to your comments “point-by-point” along with a revised version of our manuscript showing the changes required in light of your comments, newly added paragraphs, amended text, and updated references using “GREEN HIGHLIGHTS”.
- We confirm that all your comments were taken carefully into consideration, and we believe that we replied to your comments to an appropriate degree.
Q1. The author is suggested to italicize the scientific name and once use the full name after this use only O. niloticus throughout the manuscript. This makes the paper more worthy.
Task completed: ✔.
- Thank you for your suggestion. The scientific name has been italicized, and Oreochromis niloticus is fully mentioned once, followed by niloticus throughout the manuscript. (Kindly see the new lines 34, 97, 131).
Q2. Such type of mistakes makes the paper poor and less interesting for readers/reviewers please remove such kind of errors throughout the paper.
Task completed: ✔.
- Thank you for your observation. We have carefully reviewed and corrected such errors throughout the manuscript. (Kindly see the new line 99).
Q3. Correct this error.
Task completed: ✔.
- Thank you for your observation. The error has been corrected. (kindly see the new lines 125).
Q4. Write the fiberglass dimension (what is the length, width, and height of these tanks) properly.
Task completed: ✔.
- Thank you for your suggestion. The dimensions of the fiberglass tanks have been properly specified. (kindly see the new lines 133).
Q5. Correct this error.
Task completed: ✔.
- Thank you for your observation. The error has been corrected. (kindly see the new lines 135).
Q6. Write number of fish/ aquarium that was added in each fiberglass tanks during trial.
Task completed: ✔.
- Thank you for your suggestion. The number of fish per aquarium has been specified accordingly. (Kindly see the new line 137).
Q7. Were these products tested for purity and potency?
Task completed: ✔.
- Thank you for your inquiry. The purity and potency of the tested products were carefully considered: Liposomal Vitamin C (LVC): The product was commercially sourced (ASIN: B00JFF48I6), and detailed specifications, including its composition and quality, are available from the manufacturer. (Please refer to Line 146.) Coenzyme Q10 (CoQ10): The product was obtained from MEPACO Company, Cairo, Egypt, with a reported purity of >98% (Line 147). Bee Venom (BV): The composition and characteristics of the collected bee venom were confirmed through GC-MS analysis, as detailed in Table 2.
Q8. What specific parameters were used for the semiautomatic collection system (e.g., voltage, pulse duration)?
Task completed: ✔.
- Thank you for your inquiry. Please check new details in lines 150-151.
Q9. How were the storage conditions for the purified bee venom maintained at 4°C?
Task completed: ✔.
- Thank you for your inquiry. The purified bee venom was stored at 4°C in an airtight, light-protected container to preserve its biochemical integrity and prevent degradation. Please check lines 154-155.
Q10. How were the nutritional requirements of Nile tilapia determined for the formulation of the basal diet?
Task completed: ✔.
- Thank you for your inquiry. The nutritional requirements of Nile tilapia (Oreochromis niloticus) for the basal diet formulation are determined based on:
- Established Guidelines – Reference to NRC (2011), FAO, and scientific literature (https://doi.org/10.1007/s10499-011-9480-6).
- Key Nutrients – Protein, lipids, carbohydrates, essential amino acids, vitamins, and minerals.
- Ingredient Selection – Common sources include fishmeal, soybean meal, corn gluten meal, fish oil, and vitamin-mineral premixes.
- Experimental Validation – Diets are tested for growth performance, feed conversion ratio, and nutrient utilization.
Jobling, M. National Research Council (NRC): Nutrient requirements of fish and shrimp. Aquaculture International 2012, 20, 601-602, https://doi.org/10.1007/s10499-011-9480-6.
- (Kindly see the new line 168).
Q11. The discussion provides a good interpretation of the results in the context of previous literature. However: Avoid excessive repetition of results in this section.
Task completed: ✔.
- Thank you for your inquiry. Feeding 3 times a day instead of twice:
- Enhances growth performance and feed conversion ratio.
- Supports juvenile fish with higher metabolic needs.
- Compensates for plant-based diets with lower digestibility.
- Aligns with research protocol for consistent energy intake.
- Daily Siphoning:
- Removes uneaten feed, feces, and debris to prevent water contamination.
- Maintains optimal water quality by controlling ammonia, pH, and oxygen levels.
- Ensures a healthy environment for fish growth and well-being.
Q12. Correct this sentence.
Task completed: ✔.
- Thank you for your observation. The error has been corrected. (kindly see the new lines 206).
Q13. For amylase and lipase activity write few lines about the procedure i.e solution preparation etc.
Task completed: ✔.
- Thank you for your comments. A brief description of the procedure, including solution preparation, has been added accordingly. (kindly see the new lines 207-209).
Q14. How was the consistency of the 5 μm thick tissue sections ensured during preparation with the rotary microtome (Leica RM2035)?
Task completed: ✔.
- Thank you for your comment. The consistency of the 5 μm thick tissue sections was ensured by regular calibration of the rotary microtome (Leica RM2035) and proper blade maintenance to achieve uniform slicing. Additionally, the paraffin-embedded tissue blocks were properly oriented and evenly trimmed before sectioning to minimize variations in thickness. Please check lines 219-223.
Q15. It is noticed that some values in this table do not have superscripts so add superscripts from SR to K.
Task completed: ✔.
- Thank you for your valuable feedback. Superscripts were not added because there were no significant differences among the values.
Q16. Correct this sentence.
Task completed: ✔.
- Thank you for your observation. The error has been corrected. (kindly see the new lines 280).
Q17. Write the scientific name it will be more decent and make paper more presentable.
Task completed: ✔.
- Thank you for your suggestions. We have included the scientific name in all tables and figure captions.
Q18. Please also label the histology slides as written below.
Task completed: ✔.
- Thank you for your suggestions. The histology slides have been labeled accordingly from D1–D4. (Kindly see Figures 6 and 7).
Q19. It is suggested to the author that many grammatical mistakes need to be corrected.
Task completed: ✔.
- Thank you for your suggestions. The grammatical errors have been reviewed and corrected accordingly.
Finally, all authors and I would like to thank you for your notice and valuable reviews, and we hope that we responded to all your comments to meet your expectations.
Reviewer 3 Report
Comments and Suggestions for Authors
This study has generally concluded that Dietary supplementation with bee venom (4 mg/kg diet), coenzyme Q10 (60 mg/kg 534 diet), and vitamin C (200 mg/kg diet) significantly enhanced growth performance, im-535 immune function, antioxidant capacity, and intestinal health in Nile tilapia over a 60-day 536 feeding trial. However, this paper should improve some aspects, such as the scientific reasons why these doses or concentrations were used.

Author Response
Response to Reviewer #3:
This study has generally concluded that Dietary supplementation with bee venom (4 mg/kg diet), coenzyme Q10 (60 mg/kg 534 diet), and vitamin C (200 mg/kg diet) significantly enhanced growth performance, im-535 immune function, antioxidant capacity, and intestinal health in Nile tilapia over a 60-day 536 feeding trial. However, this paper should improve some aspects, such as the scientific reasons why these doses or concentrations were used.
Response: -
- All authors thank the reviewer for their valuable comments and constructive feedback, which have significantly improved the quality of our manuscript. We have addressed each comment in detail and provided a point-by-point response, along with a revised version of the manuscript. Changes, including newly added paragraphs, amended text, and updated references, are highlighted in Turquoise.
- We confirm that we have carefully considered all your comments and believe our responses adequately address the points raised.
Q1. Line 34 Oreochromis niloticus should be Italic
Task completed: ✔.
- Thank you for your insightful comment. The scientific name has been italicized. (Kindly see the new line 34).
Q2. Line 38 The feeding rate should be mentioned in this sentence.
Task completed: ✔.
- Thank you for your suggestion. The feeding rate has been included accordingly. (Kindly see the new lines 37-38).
Q3. Line 135 Are the fish fasted during the 14 days of acclimatization? Please clarify and complete the information on what was done during acclimatization.
Task completed: ✔.
- Thank you for your inquiry. The details regarding the acclimatization period have been clarified accordingly. (Kindly see the new line 135).
Q4. Lines 144-150 How to harvest bee venom should be shown in visual form.
Task completed: ✔.
- Thank you for your inquiry. As the volatile components naturally evaporated, the venom solidified into a white precipitate, which was carefully scraped off the glass surface for collection. Please check lines 152-154.
Q5. Line 160 It would be better to include a scientific rationale for using test concentrations for liposomal Vitamin C (200 mg/kg), CoQ10 (60 mg/kg), and BV 160 (4 mg/kg) in this study.
Task completed: ✔.
- Thank you for your insightful question. The additive levels were selected based on previous studies that have recommended these specific concentrations. The relevant references supporting our choice are as follows: (https://doi.org/10.1007/s10499-025-01882-4; https://doi.org/10.1016/j.fsi.2020.01.052; https://doi.org/10.1016/j.fsi.2024.109713). (kindly see the new lines 165-167).
Q6. Lines 310 & 318 Please increase the font size on the chord diagram and also the PCA to make it easier to read.
Task completed: ✔.
- Thank you for your feedback. We have improved the clarity and resolution of Figures 2 and 3 to enhance their readability. Please review the revised figures in the updated manuscript.
Q7. Lines 385, 386, 402. 403 There's no need to show the inside thick on all the graphs in Figure 8 and Figure 9.
Task completed: ✔.
- Done as requested.
Finally, all authors and I would like to thank you for your notice and valuable reviews, and we hope that we responded to all your comments to meet your expectations.
Round 2
Reviewer 1 Report
Comments and Suggestions for Authors
The authors have revised the manuscript according to the previous comments. However, the formula of WGR should be re-corrected.
Author Response
Response to Reviewer #1: -
The authors have revised the manuscript according to the previous comments. However, the formula of WGR should be re-corrected.
Response: -
- All authors thank the reviewer for his/her valuable comments and constructive reviews for improving our manuscript. Please check the corrected formula of WGR% in line 187.
Finally, all authors and I would like to thank you for your notice and valuable reviews, and we hope that we responded to all your comments to meet your expectations.
